# Control of Polarization Orientation Angle of Scattered Light Based on Metasurfaces: −90° to +90° Linear Variation

**DOI:** 10.3390/ma15062076

**Published:** 2022-03-11

**Authors:** Song Wu

**Affiliations:** College of Chip Industry, Hubei University of Technology, Wuhan 430068, China; 20191049@hbut.edu.cn

**Keywords:** linear to linar polarization, linear to circular polarization, polarizer, polarization manipulation

## Abstract

Metasurfaces can be used to precisely control polarization state of the scattered light. Here, we present a metasurface-based terahertz device. On the one hand, it serves as a high performance linear polarization converter in transmission of over 80% power with weak reflection. It is capable of rotating linear polarization orientation angle with respect to *x*-axis continuously from −90° to +90° at 0.84 THz. On the other hand, it serves as a circular polarizer. It can transform a linear polarized wave into a circular polarized wave at 2.49 THz. The transmitted and reflected field are both circular polarized with 50% power. The proposed device with dual functionalities can be applied to modulate the polarization state of the signal in THz wireless communication.

## 1. Introduction

The terahertz frequency range (0.1–10 THz) could form the basis of the next generation of high-speed wireless communication, as well as a range of spectroscopy, sensing, and imaging applications [1,2,3,4]. Spatiotemporal control of the far- and near-field distribution of terahertz signals, and the manipulation of terahertz fields will be crucial in enabling these applications.

Metasurfaces are two-dimensional surfaces with precisely designed scatterers that create controlled field transformation of incident wavefronts across the properties of amplitude [5,6], phase [7,8,9], frequency [10,11], and polarization [12,13,14,15]. Polarization is one of the basic properties of electromagnetic waves conveying valuable information in signal transmission and sensitive measurements. Cross-polarization conversion has been demonstrated using 2D vertical-shape metasurface [16,17]. V-shape metasurfaces with different sizes are arranged together to transform linearly polarized waves into circularly polarized waves and work for any orientation of the incident linear polarization [18]. Rod-shape metasurface and a metal ground plane form a Fabry-Pérot-like cavity to result in cross-polarized scattering in reflection [19]. Stacked rod-shaped arrays with a tailored rotational twist can manipulate circular polarization in broadband spectrum [9,20]. However, the currently available designs suffer from lack of continuous modulation and low effective power.

In this article, we report a metasurface comprised of circular-shape split-ring resonator (SRR) that offer continuous adjustment of polarization orientation angle and realize linear to circular polarization. Circular-shape SRR, as shown in Figure 1a, is simultaneous control of phase and amplitude of the incident light, which are separately controlled by opening angle α and rotation angle, γ. All models and simulated calculations are carried out in CST Microwave Studio 2020, and circular-shape SRR is made of Au. For a linearly polarized incident beam, the polarization orientation angle θ is continuously varied from −90° to +90° by controlling the rotation angle γ at 0.84 THz and the incident linear polarization is transformed into circular polarization by controlling the opening angle α at 2.49 THz. High effective power of 80% is achieved by means of stacking circular-shape SRR.

## 2. Linear Polarization Orientation Angle Modulation

We choose incident light ${\vec{E}}_{in}(z,t)=cos(2\pi ft+kz){\vec{e}}_{x}$ propagating along-*z*-axis direction, where *f* is the frequency of the incident light, *t* is the time, and *k* is the wavenumber, whereas the scattered light has a component which is polarized orthogonal to that of the incident light and excited by the symmetric and antisymmetric current modes [6,17,21]. The scattered light can be expressed as:(1)$${\vec{E}}_{scat}(z,t)=E_{x}(z,t){\vec{e}}_{x}+E_{y}(z,t){\vec{e}}_{y}$$
where $E_{x}(z,t)=\left|t_{xx}\right|\cos\left(2\pi ft+kz+\varphi_{xx}\right)$, $E_{y}(z,t)=\left|t_{yx}\right|\cos\left(2\pi ft+kz+\varphi_{yx}\right)$. $t_{xx}=\left|t_{xx}\right|e^{j\varphi_{xx}}$ and $t_{yx}=\left|t_{yx}\right|e^{j\varphi_{yx}}$ are the transmission coefficient for *x*- and *y*-components, respectively. The scattered wave is linear polarized wave if $\varphi_{xx}-\varphi_{yx}=0^{{}^{\circ}},\pm 180^{{}^{\circ}}$. Polarization orientation angle is $\theta =\arctan\left(\left|t_{yx}\right|/\left|t_{xx}\right|\right)$. The scattered power of interest is determined by the amplitude of the scattered wave, ${\left|{\vec{E}}_{scat}\right|}^{2}={\left|t_{xx}\right|}^{2}+{\left|t_{yx}\right|}^{2}$. Circular-shape SRR metasurface is able to change transmission coefficient continuously and meet the phase condition by rotating γ. Figure 1b,c show the simulated transmission amplitude and phase variations of an array of circular-shape SRR at different γ from −45° to +45° at 0.84 THz. $\left|t_{yx}\right|$ is approximate to vary as sin(2θ)/2 while the phase remains constant in two separated ranges of −45° to 0° and 0° to +45°, with an abrupt change of 180° at $\gamma =0^{{}^{\circ}}$. $\left|t_{xx}\right|$ follows a cos(4θ/3) dependence while the phase remains constant. Thus, the rotation angle γ serves as one important parameter to control the amplitude of a scattered wave without resorting to new geometry.

Figure 2a–g show the variation of electric field of scattered light in time domain at *z* = −10 µm. The scattered light is linearly polarized along *x*-axis with polarization orientation angle θ varying from −45° to +45°. The polarization orientation angle θ is proportional to the rotation angle γ, θ/γ≈1. The larger polarization orientation angle is, the lower amplitude and effective power of the scattered light is, as is shown in Figure 2h. The scattered power of interest is higher than 80% at $\theta =0^{{}^{\circ}}$ with low reflections (Figure 1d) and less than 50% at $\theta =\pm 45^{{}^{\circ}}$ with high reflections (Figure 1d). So, the single-layered metasurface produced weak scattered beams with the increase of γ and with increased reflections.

In order to improve the power of scattered light in transmission with reduced reflections, we use two metasurfaces to result in much improved performance. An example consisting of two-separated circular-shape SRRs is shown in Figure 3a, which are separated by air of thickness *d* = 50 µm. This corresponds to a separation of $\lambda_{0}/8$ at 0.84 THz. The rotation angle of layer 1 is set as $\gamma =0^{{}^{\circ}}$.

The scattered light from two-layered circular-shape SRRs is linearly polarized with respect *x*-axis. Polarization orientation angle is controlled by rotation angle $\gamma_{1}$ of layer 2. The electric field vector of the scattered light is shown in Figure 3b with $\gamma_{1}$ varying from −45° to +45°. The polarization orientation angle of two-layered circular-shape SRRs (blue solid line in Figure 3b) is continuously varied from −45° to +45° with the same scattered power of more than 80% at 0.84 THz. Compared with the single-layer circular-shape SRR (red solid line in Figure 3b), two-layer circular-shape SRRs have better impedance matching to reduce reflections with the increase of $\gamma_{1}$, so the scattered power of interest is enhanced. So far, continuous polarization orientation angle modulation from −45° to +45° with high effective power of more than 80% is achieved by stacking two-layer circular-shaped SRRs. However, two-layered structure cannot result in polarization orientation angle of −90° to −45° and +45° to +90° with effective power of more than 80%. When $\gamma_{1}$ exceeds $\pm 45^{{}^{\circ}}$, the scattered power of interest decreases significantly. One way to overcome this problem is to use a three-layer structure.

Figure 4b,c show two different styles for three-layer structures. Style Ⅰ is used to continuously control polarization orientation angle from +45° to +90° by rotating $\gamma_{2}$ of layer 3 from +45° to +90° (black solid line in Figure 4a). Style II is used for control of polarization orientation angle from −45° to −90° by varying $\gamma_{2}$ of layer 3 from −45° to −90° (red solid line in Figure 4a). The scattered power of interest remains constant of more than 80%. Figure 5 summarizes the simulated results of linear polarization orientation angle modulation by using circular-shape SRR. Linear polarization orientation angle with respect to *x*-axis is modulated continuously from −90° to +90° at 0.84 THz and high effective power of more than 80% are obtained for different polarization orientation angle.

## 3. Circular Polarization Modulation

In the above section, linear polarization orientation angle modulation is demonstrated by the amplitude transformation ability through circular-shape SRR metasurface. $\left|t_{xx}\right|$ and $\left|t_{yx}\right|$ are continuously controlled by rotation angle γ while maintaining almost the same phase or out of phase. In this section, the phase transformation ability through circular-shape SRR metasurface is demonstrated by condsidering circular polarization. We choose the incident polarization to be at 45° with respect to *x*-axis and ${\vec{E}}_{\mathrm{in}}$ can be decomposed into two components, a vertical component ${\vec{E}}_{⊥}$ and a horizontal component ${\vec{E}}_{∥}$, as is shown in Figure 6a. $t_{⊥}=\left|t_{⊥}\right|e^{j\varphi_{⊥}}$ and $t_{∥}=\left|t_{∥}\right|e^{j\varphi_{∥}}$ are transmission coefficients for vertical and horizontal component respectively. $\varphi_{⊥}$ and $\varphi_{∥}$ are continuously controlled by opening angle α while maintaining similar amplitude, as is shown in Figure 6b,c. When $\alpha =55.5^{{}^{\circ}}$, $\varphi_{⊥}$−$\varphi_{∥}$ will be −90° at 2.49 THz. The scattered light is a circular polarization wave with effective power of 50%, as is shown in Figure 6d. Another 50% of the power is reflected in circularly polarized waves [22]. So the transmitted and reflected wave share the same power and wave form.

In order to improve effective power of interest by decreasing the reflected power, the two-layered circular-shape SRR circular polarizer is designed, as is shown in Figure 7a. Two-layer circular-shape SRRs are the same geometry and spaced by air of thickness *d*_1_ = 42.3 µm. The scattered light is circular polarized at 2.07 THz (blue solid line in Figure 7b). Compared with a single layer polarizer (red solid line in Figure 7b), two-layered polarizer has an effective power of more than 80% with reduced reflections, which can be seen from Figure 7b. The amplitude of the transmitted circularly polarized wave is larger than 80%.

## 4. Conclusions

We have reported a circular-shape SRR with amplitude and phase transformation ability. The rotation angle γ can control the scattered amplitude while maintaining almost the same phase. The opening angle α can control the scattered phase while maintaining a similar amplitude. Continuous linear polarization orientation angle modulation from −90° to +90° is realized by the amplitude transformation ability. Linear to circular polarization modulation is realized by the phase transformation ability. In order to improve the effective power of interest, we use a two- or three-layer structure to result in effective power up to 80%.

## Figures and Tables

**Figure 1 materials-15-02076-f001:**
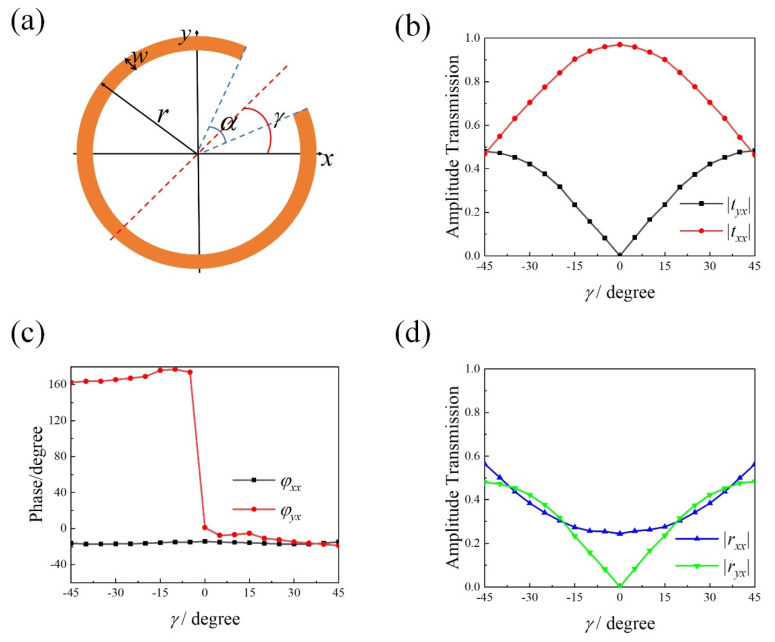
(**a**) Schematic of circular-shaped SRR with opening angle α, radius r, width w and rotation angle θ with respect to *x*-axis. (**b**) Simulated transmission amplitude and (**c**) phase of an array of circular-shape SRR with (r, α, w) = (34 µm, 45°, 5 µm) and γ varying from −45° to 45° at 0.84 THz. (**d**) Simulated reflection amplitude. The period of the array is 80 µm in both *x*- and *y*-direction.

**Figure 2 materials-15-02076-f002:**
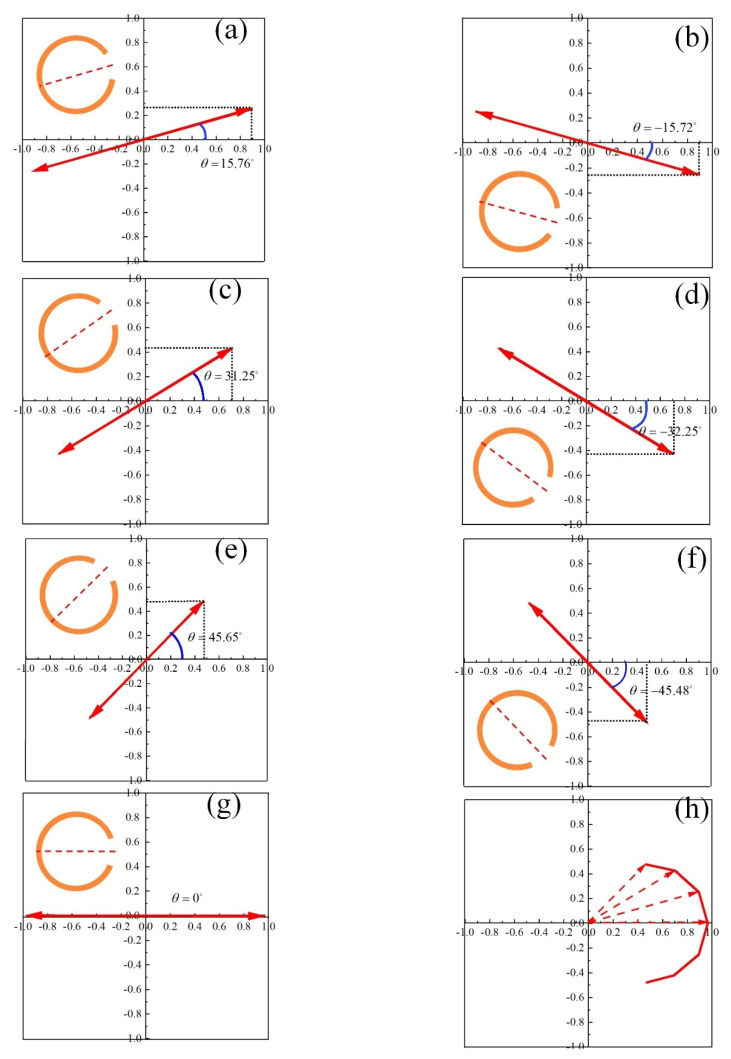
Variation of electric field of scattered light in time domain at *z* = −10 µm for (**a**) $\gamma =+15^{{}^{\circ}}$ and (**b**) $\gamma =-15^{{}^{\circ}}$, (**c**) $\gamma =+30^{{}^{\circ}}$ and (**d**) $\gamma =-30^{{}^{\circ}}$, (**e**) $\gamma =+45^{{}^{\circ}}$ and (**f**) $\gamma =-45^{{}^{\circ}}$, (**g**) $\gamma =0^{{}^{\circ}}$. (**h**) Electric field vector of the scattered light with γ varying from −45° to +45° at 0.84 THz.

**Figure 3 materials-15-02076-f003:**
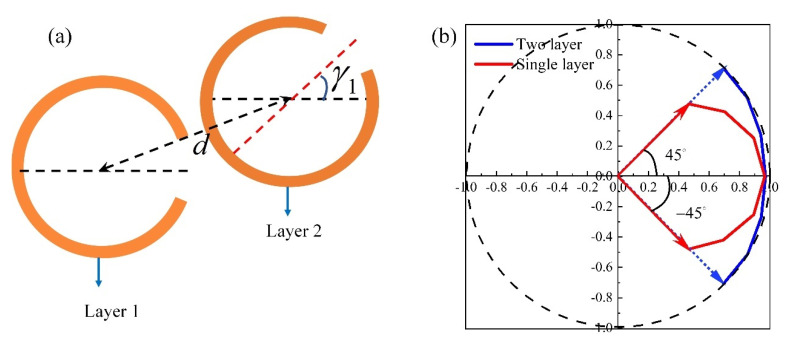
(**a**) Two-layer circular-shape SRR. (**b**) Electric field vector variation of the scattered light at 0.84 THz with rotation angle $\gamma_{1}$ varying from −45° to +45°, where the red and blue line represent the single-layer and two-layer metasurface, respectively.

**Figure 4 materials-15-02076-f004:**
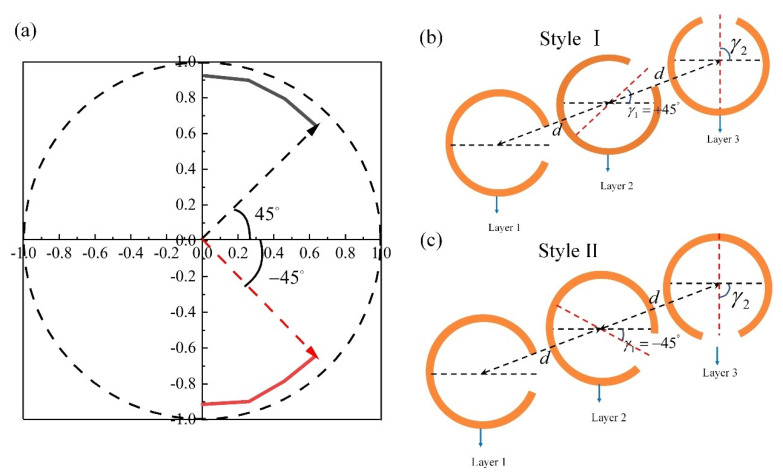
(**a**) Electric field vector of the scattered light at 0.84 THz with rotation angle $\gamma_{2}$ varying from +45° to +90° (red solid line) and −45° to −90° (black solid line). Schematic of the three-layer structure: (**b**) Style Ⅰ with $\gamma_{1}=+45^{{}^{\circ}}$ and (**c**) Style II with $\gamma_{1}=-45^{{}^{\circ}}$.

**Figure 5 materials-15-02076-f005:**
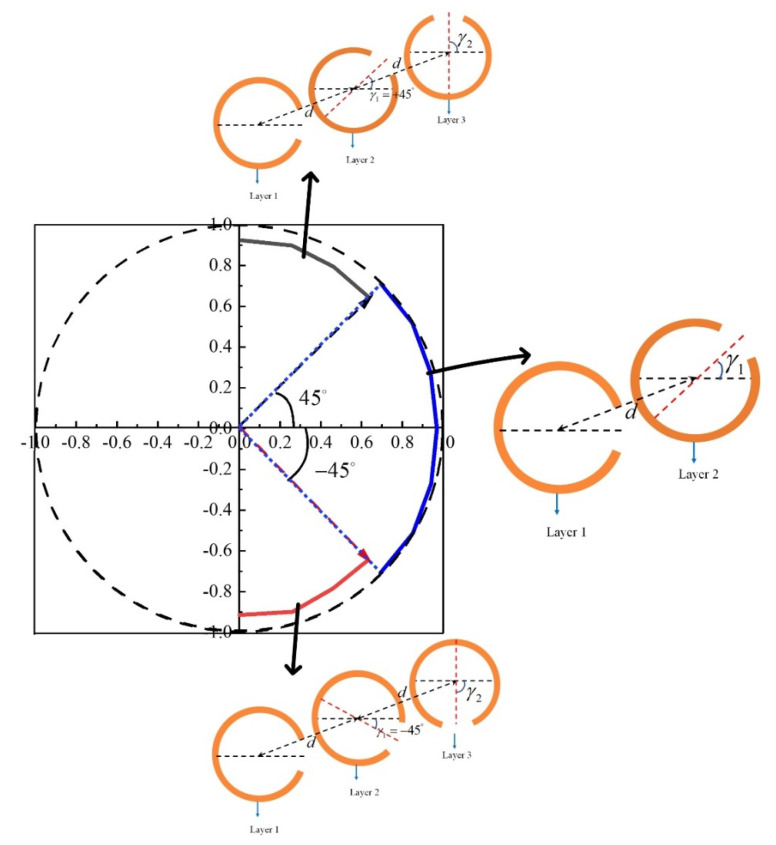
Linear polarization orientation angle modulation and its corresponding circular−shape SRR structure.

**Figure 6 materials-15-02076-f006:**
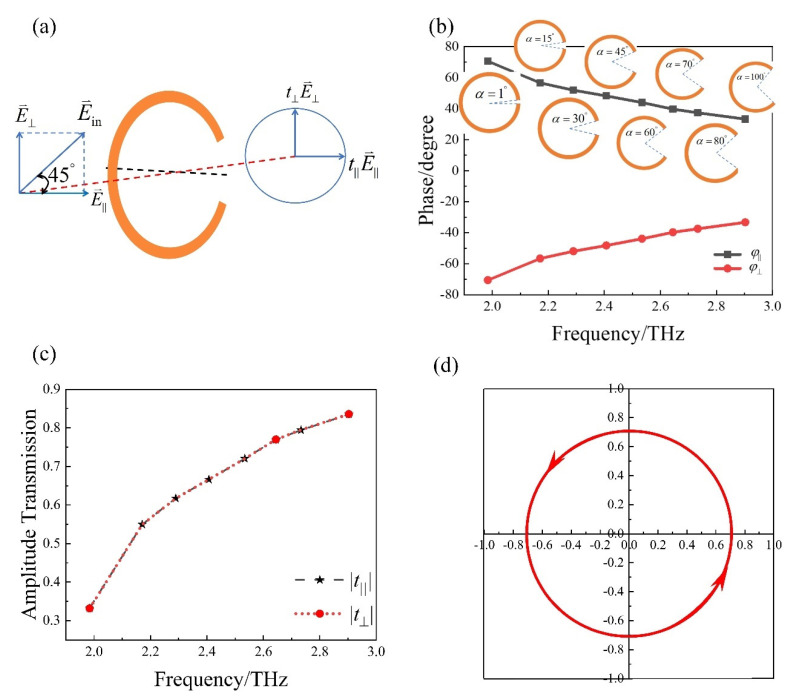
(**a**) Schematic of circular−shaped SRR to realize linear to circular polarization with rotation angle $\gamma =0^{{}^{\circ}}$. Simulated transmission (**b**) phase and (**c**) amplitude for vertical and horizontal components with opening angle α varying from 1° to 100°. (**d**) Variation of electric field of scattered light in time domain at 2.49 THz, at z = −10 µm.

**Figure 7 materials-15-02076-f007:**
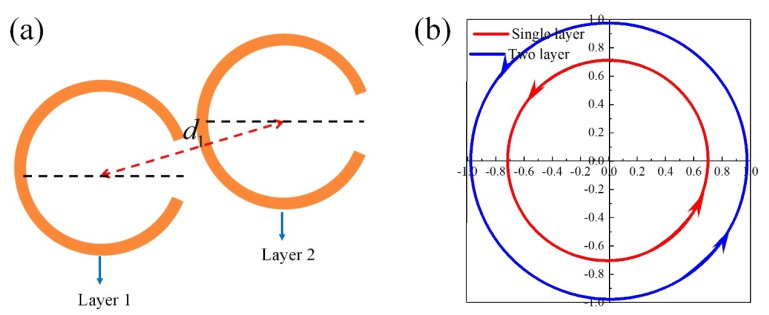
(**a**) Schematic of circular-shape SRR circular polarizer with (r, α, w, γ) = (39 µm, 60°, 1 µm, 0°) and $d_{1}=42.3\;\mu m$. (**b**) Variation of electric field of scattered light in time domain at z = −10 µm, where the red and blue line represent the single−layer and two-layered circular-shape SRR polarizers, respectively.

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
