# Peer review of "Control of Polarization Orientation Angle of Scattered Light Based on Metasurfaces: −90° to +90° Linear Variation"

_materials, 2022, doi:10.3390/ma15062076_

Round 1

Reviewer 1 Report

This paper calls: "Metasurfaces with control of polarization orientation angle of scattered light -90° to +90° linear variation" and concerned of actual topic of research - metasurfaces with special properties of state polarization controling. The advantage of this article is actual theme of research and good references review. The disadvantage of this article is absence of experimental part of article.

Author Response

Thanks for affirming the value of this paper. The work of this paper is to put forward the theoretical part. Our laboratory still lacks relevant experimental equipment, so it is hard for me to give a experimental results. But I think the experimental results are very important.

Reviewer 2 Report

The authors presents a work the investigation of the control of polarization state of scattered light by means of metasurfaces. The paper is interesting and the information given by the authors are well supported by the bibliography. The paper in its current form could be accepted for pubblication after minor revision due to the presence of many inaccuracies and errors such as the correction of some English typos and high-definition figures, and the addition of some information that can help the reader.
In the attached file I propose different comments/corrections to be made in the article, referring to the proof copy.

Author Response

Thanks for your advices. I have made a comprehensive revision to the draft according to your comments. Please see the attachment.

Reviewer 3 Report

Fascinating work, with wide applicability! A slightly rewritten paper would explain these questions I have:

The results are entirely a computer simulation, right? If not, include a diagram of the experimental setup.

How do you make these structures?

Figure 6: Do you mean electric field amplitude, or intensity transmision. If amplitude, total transmission is ~65%. 

The last sentence is interesting: is it true that you need a 2 or 3 layer structure to get 80% transmission? If so, how would you make that?

C. Phipps

Author Response

Thanks for your advices. I have made a comprehensive revision to the draft according to your comments. Please see the attachment

Round 2

Reviewer 1 Report

The author answered on all questions and article can be published in present form.